# Stratified Effects of Tillage and Crop Rotations on Soil Microbes in Carbon and Nitrogen Cycles at Different Soil Depths in Long-Term Corn, Soybean, and Wheat Cultivation

**DOI:** 10.3390/microorganisms12081635

**Published:** 2024-08-10

**Authors:** Yichao Shi, Alison Claire Gahagan, Malcolm J. Morrison, Edward Gregorich, David R. Lapen, Wen Chen

**Affiliations:** 1Ottawa Research and Development Centre, Agriculture and Agri-Food Canada, 960 Carling Ave., Ottawa, ON K1A 0C6, Canada; yichao.shi@agr.gc.ca (Y.S.); claire.gahagan@agr.gc.ca (A.C.G.); malcom.morrison@agr.gc.ca (M.J.M.); edward.gregorich@agr.gc.ca (E.G.); david.lapen@agr.gc.ca (D.R.L.); 2Department of Biology, University of Ottawa, 60 Marie Curie Prv., Ottawa, ON K1N 6N5, Canada

**Keywords:** no-till, crop rotation, bacterial communities, carbon fixation, nitrogen, PICRUSt2

## Abstract

Understanding the soil bacterial communities involved in carbon (C) and nitrogen (N) cycling can inform beneficial tillage and crop rotation practices for sustainability and crop production. This study evaluated soil bacterial diversity, compositional structure, and functions associated with C-N cycling at two soil depths (0–15 cm and 15–30 cm) under long-term tillage (conventional tillage [CT] and no-till [NT]) and crop rotation (monocultures of corn, soybean, and wheat and corn–soybean–wheat rotation) systems. The soil microbial communities were characterized by metabarcoding the 16S rRNA gene V4–V5 regions using Illumina MiSeq. The results showed that long-term NT reduced the soil bacterial diversity at 15–30 cm compared to CT, while no significant differences were found at 0–15 cm. The bacterial communities differed significantly at the two soil depths under NT but not under CT. Notably, over 70% of the tillage-responding KEGG orthologs (KOs) associated with C fixation (primarily in the reductive citric acid cycle) were more abundant under NT than under CT at both depths. The tillage practices significantly affected bacteria involved in biological nitrogen (N_2_) fixation at the 0–15 cm soil depth, as well as bacteria involved in denitrification at both soil depths. The crop type and rotation regimes had limited effects on bacterial diversity and structure but significantly affected specific C-N-cycling genes. For instance, three KOs associated with the Calvin–Benson cycle for C fixation and four KOs related to various N-cycling processes were more abundant in the soil of wheat than in that of corn or soybean. These findings indicate that the long-term tillage practices had a greater influence than crop rotation on the soil bacterial communities, particularly in the C- and N-cycling processes. Integrated management practices that consider the combined effects of tillage, crop rotation, and crop types on soil bacterial functional groups are essential for sustainable agriculture.

## 1. Introduction

Soil-borne bacterial communities play pivotal roles in fundamental ecological processes, including soil formation, nutrient cycling, and organic matter turnover [1,2]. Their significance in agroecosystem sustainability and multifunctionality, particularly in carbon (C) and nitrogen (N) cycling, is well established [2,3]. Understanding the diversity, composition, and functions of bacterial communities is crucial for unraveling the intricate mechanisms that drive soil health and crop productivity and thus imperative to inform agricultural management decisions.

In agricultural systems, tillage and crop rotation significantly influence soil physicochemical properties, subsequently shaping the diversity and composition of the soil microbiota [4,5,6,7,8]. Conventional tillage (CT) practice disrupts soil aggregates and microbial inhabitants [9], while no-till (NT) improves soil structure, moisture retention, and organic matter content [10], promoting bacterial diversity and often increasing crop yields [11,12,13]. However, NT may constrain soil aeration in heavier soils [14]. Additionally, the tillage effects on soil bacterial community diversity and composition vary with soil depth [15,16]. For example, Sun et al. [7] found that tillage with a moldboard plow reduced the variability in bacterial community structure across soil depths compared to NT practices. Li et al. [15] demonstrated that the alpha diversity indices of bacterial communities differed significantly between the 0–5 cm and the 5–20 cm soil depths under NT practices but not under CT. Higher community structural variance was also observed between these two depths under NT compared to CT. These observations highlight the stratified effects of tillage practices on microbial functional guilds relevant to soil fertility and crop production.

Crop rotation manages soil nutrients, water availability, weeds, and pests [17]. Using crop rotation schemes that include both C3 and C4 plants, such as corn (*Zea mays* L.)–soybean (*Glycine max* L. Merr.)–wheat (*Triticum aestivum* L.) rotations, offers numerous benefits for agricultural sustainability and productivity [18]. C3 crops, like soybean and wheat, often improve soil nitrogen levels and enhance soil structure in the top layers, while C4 crops, like corn, use water and nutrients more efficiently and thrive in hotter, drier conditions [19]. This complementary relationship helps enhance C sequestration and nutrient use efficiency, breaks pest and disease cycles, and promotes diverse root systems, thus improving soil health and reducing erosion [20,21]. Rotations between leguminous and gramineous crops coupled with stubble management strategies facilitate the transfer of organic matter and nutrients (especially N) from legumes to the subsequent crop through the rhizodeposition effect [22,23]. Crop rotation has shown to enrich N_2_-fixing and growth-promoting bacteria in the surface soil layer [24,25] and along the entire rooting depth through root exudates [26,27]. These effects were evident in bean–wheat rotations [28] and in cropping systems involving alfalfa, bean, and clover [29]. By contrast, microbial communities in sub-surface soils are likely more associated with long-term C sequestration due to the characteristics of subsoil organic matter resulting from microbial decomposition [30]. Understanding the role of crop rotation in modifying bacterial community diversity and function across soil depths provides valuable insights into sustainable agricultural strategies to ensure soil health and productivity.

While numerous studies have explored the impact of tillage practices and crop rotation regimes on the soil microbiome, there is a need for further investigation into the ecological functions of specific microbial communities. This is due to their critical role in regulating nutrient cycling and greenhouse gas emissions, areas where the effects of agricultural management are significant, yet not fully understood [31,32]. Previous studies have investigated the effects of agricultural practices on soil C fixation by focusing on the *cbbL* gene, which governs the Calvin–Benson cycle. These studies showed that *cbbL*-carrying bacteria were less abundant under CT than under NT practices [33] and that the tillage practices substantially influenced their compositional structure across different soil aggregate sizes [34]. Despite progress, understanding remains limited in how soil bacteria, beyond those involved in the Calvin–Benson cycle, respond to different agricultural practices. This gap extends to other C fixation pathways. Additionally, N cycling, a pivotal biogeochemical process in agroecosystems, is predominantly mediated by microorganisms [35]. Exploring the response of the soil bacterial communities involved in various C- and N-cycling processes to tillage and crop rotation practices will enhance our understanding of nutrient cycling and the sustainability of agroecosystems.

Advancements in high-throughput sequencing technologies have revolutionized our ability to explore environmental microbial communities using metagenomics approaches, including metabarcoding, shotgun metagenomics, and metatranscriptomics [36]. Among these, metabarcoding stands out for its rapid, cost-effective, and comprehensive analysis of mixed-specimen DNA fragments [37]. This technique enables the assessment of microbial diversity and abundance within ecosystems, providing deep insights into ecological dynamics [37,38]. Although primarily used for biodiversity discovery, functional prediction tools, like PICRUSt2 [39,40] and Tax4Fun2 [41], can infer the functional potential of bacterial communities from 16S rRNA gene metabarcodes. These tools offer insights into the coding potential of community genomes, although these predictions are still considered prospective for certain genes [39,40]. Recent studies leveraging PICRUSt2 have effectively explored soil bacterial functions related to C and N cycling, highlighting the significant impact of tillage on the microbial communities involved in these biological processes [6,42].

Long-term agricultural experimental sites play a crucial role in assessing the impact of agronomic practices on crop productivity, soil quality, and associated microbial processes [43]. Building on this understanding, in 1990, Agriculture and Agri-Food Canada (AAFC) established a long-term study site at the Ottawa Research and Development Centre’s Centre Experimental Farm, Ottawa, Ontario, Canada, to evaluate the effects of tillage and rotation practices on soil health and crop productivity [44]. Specifically, the crop rotation regime involves corn, soybean, and wheat [44,45], which are central to legume–cereal rotations in Eastern Canada and other regions across North America, underlining the site’s relevance to broader agricultural practices. The current study aimed to assess the effects of tillage and crop rotation regimes on the diversity and compositional structure of soil bacterial communities utilizing 16S rRNA gene amplicon-based Illumina sequencing. Particularly, by utilizing bioinformatics tools like PICRUSt2, this study evaluated bacterial functions related to C fixation and N cycling across the soil profile at 0–15 cm and 15–30 cm depths.

## 2. Materials and Methods

### 2.1. Study Site and Experimental Design

Soil samples for the study were collected from a tillage and crop rotation trial established in 1990, situated at the Central Experimental Farm of Agriculture and Agri-Food Canada (AAFC)’s Ottawa Research and Development Centre, Ottawa, ON, Canada (45°23′13.6″ N; 75°43′15.6″ W) [44,45]. The soil type was Matilda sandy loam (Melanic Brunisol according to Canadian classifications) with a pH (in CaCl_2_) of 6.8 [44]. The experimental design, consisting of a split-plot arrangement with two replicates, aimed at exploring the interactions between tillage practices and crop rotation regimes. The main plots (89.1 m × 45.7 m each) underwent either no-tillage (NT) or conventionally tillage (CT) treatments. CT was performed with a moldboard plough (to ~20 cm depth) in early November, followed by spring cultivation using a mulch finisher and a combination harrow with rotatory baskets. Nested within these main plots, subplots of 9.1 m by 45.7 m were established for different crop rotation schemes. These subplots featured three crops (corn [C], soybean [S], or wheat [W]) either in monoculture (CCC, SSS, CCC) or in two 3-year rotation sequences: corn–soybean–wheat [CSW] or corn–wheat–soybean [CWS]. Each rotation sequence was initiated so that all three crops were planted annually, resulting in nine subplots per main plot: three for CSW, three for CWS, and three for each monoculture. To assess the tillage effects, these crop and rotation combinations were replicated as complete blocks within each main plot. Consequently, the experiment comprised 72 subplots in total, with these randomized and replicated to cover nine rotation sequences (*n* = 2 × 9) across both sets of duplicated main plots (*n* = 2 × 2), ensuring a detailed analysis of the interactions between tillage practices, crop, and rotation strategies.

Crop planting and management information were described in detail by Morrison et al. [44] and Gahagan et al. [45]. In brief, wheat was sown in the first two weeks of May (450 seeds m^−2^ in 19 cm wide rows). Corn was seeded (7 seeds m^−2^ in 76 cm wide rows) in the first two weeks of May. Soybean was planted (55 seeds m^−2^ in 19 cm wide rows) in the last two weeks of May. Corn received a pre-plant application of 224 kg ha^−1^ of N (as urea) and an at-seeding application of 40 kg ha^−1^ of N–P_2_O_5_–K_2_O (18-18-18). Wheat was fertilized with 100 kg ha^−1^ of N (as urea) before planting, while soybean received no fertilizer. Pre-plant fertilizers were applied before spring tillage; thus, they were integrated into the soil in the CT plots but remained on the surface in the NT plots. In the CT plots, all crop residues were incorporated into the soil each fall, whereas, in the NT plots, residues were left on the surface.

From 1990 to the present, the experimental setup underwent two modifications. Specifically, after two complete cropping cycles (6 years) the tillage treatments were swapped, and the experiment focused on the effects of changing tillage on the soil. In 2001, the tillage treatment changed back to the original design and remained that way until the end of this experiment.

### 2.2. Soil Sampling

This study explored the effects of tillage practices (CT vs. NT) and crop rotation regimes (corn–soybean–wheat rotation [CSW] and monocultures of corn [C], soybean [S], and wheat [W]) on the soil microbiome at two soil depths. In 2018, soil samples were collected from the plots after harvest at the soil depths of 0–15 cm and 15–30 cm using a soil core sampler (diameter = 2 cm, Lamotte, Chesterton, MD, USA). For each plot and soil depth, composite samples were created by homogenizing 15 cores sampled in a random stagger pattern from the plot. To prevent cross-contamination, all sampling equipment was thoroughly rinsed with distilled water, sterilized with 90% ethanol, and dried prior to use on a different plot. With crop rotations, tillage practice, and replication there was a total of 96 samples. These soil samples were stored in a cold room (4 °C) until sampling was completed. The soil samples were then thoroughly mixed and sieved through a 5 mm mesh to remove rocks, plant residues, and insects. A subsample was stored at −80 °C in a 15 mL Falcon tube for DNA extraction. Another set of soil subsamples was allocated for soil physicochemical analysis. To determine the soil moisture content, a fresh 30 mL soil sample was weighed, then dried at 60 °C until reaching a stable weight (for approximately 7 days), with moisture content calculated based on the dry weight. The soil pH was measured using a 1:2 soil-to-water ratio in distilled water [46]. For the analysis of total soil C and N, the soil was sieved to less than 0.2 mm. The total soil C and N levels were quantified using a LECO CNS-1000 analyzer (LECO Corp., St. Joseph, MI, USA). The available phosphorus content was determined using the Olsen P method [47], and the available potassium was measured using the ammonium acetate extraction method [48].

### 2.3. DNA Extraction

The DNA was extracted using the FastDNA™ Spin Kit for Soil (MP Biomedicals, Santa Ana, CA, USA) following the manufacturer’s instructions, with a few modifications as previously described [45]. The DNA samples were stored at −25 °C. The concentration of the DNA extracts was measured using the Qubit dsDNA HS (High Sensitivity) Assay Kit on a Qubit Fluorometer (Thermo Fisher Scientific, Waltham, MA, USA). The plates containing the genomic DNA were shipped overnight on dry ice to the Génome Québec Innovation Centre (Montreal, QC, Canada) for the preparation of sequencing libraries and amplicon-based metagenomics (or metabarcoding) sequencing using the Illumina MiSeq platform (Illumina, San Diego, CA, USA). The target fragment length was 100–300 bp, with a target output of 15 Gb.

### 2.4. Sequencing Library Preparation and Illumina MiSeq Sequencing

At the Génome Québec Innovation Centre, amplicon-based sequencing libraries were prepared using the following protocol. The 16S rRNA gene (targeting the V4–-V5 hyper-variable region) was amplified via PCR using the gene-specific primer pair 515F-Y (5′-GTG YCA GCM GCC GCG GTA A-3′) as the forward primer and 926R (5′-CGY CAA TTY MTT TRA GTT T-3′) as the reverse primer. The initial denaturation was conducted at 95 °C for 3 min, followed by 25 cycles of denaturation at 95 °C for 30 s, annealing at 55 °C for 30 s, and extension at 72 °C, with a final extension at 72 °C for 5 min. The amplicons were verified on a 2% agarose gel, quantified, and purified using the sparQ PurMag Beads (Quantabio, Beverly, MA, USA). A secondary PCR was performed to attach dual indexed barcoding adapters. The PCR protocol included an initial denaturation at 96 °C for 15 min, followed by 30 cycles at 96 °C for 30 s, 52 °C for 30 s, and 72 °C for 60 s, with a final extension at 72 °C for 10 min. For indexing, 1 μL of undiluted amplicon product was used. The indexed samples were verified on a 2% agarose gel and quantified using the Quant-iT™ PicoGreen^®^ dsDNA Assay Kit (Life Technologies, Carlsbad, CA, USA). The sequencing library was made with an equal quantity in ng of DNA for each sample. The library DNA was cleaned with sparQ PureMag Beads. The library was quantified using Kapa Illumina GA with the Revised Primers-SYBR Fast Universal Kit (Kapa Biosystems, Wilmington, MA, USA), and the average fragment size was determined using the LabChip GX (PerkinElmer, Waltham, MA, USA). The DNA samples were stored at –80 °C before and after the sequencing library preparation. Sequencing was performed on the Illumina MiSeq platform with the MiSeq Reagent Kit v3 with 600 cycles.

### 2.5. Metabarcoding Sequencing Data Processing and Analysis

The Illumina MiSeq sequencing adapters were removed from the fastq files using Cutadapt ver. 4.1 [49]. The paired-end raw reads were processed using DADA2 ver. 1.14 [50] implemented in QIIME2 for denoising, chimera detection, and amplicon sequence variant (ASV) inference using default parameters. The raw forward and reverse reads were truncated at 200 nt. The taxonomic assignment was performed by training the Naive Bayes classifier q2-feature-classifier [51] using the Greengene2 reference database [52] with a minimum bootstrap confidence at 80%. Functional annotation of the ASV table was inferred using PICRUSt2 [39]. The abundance table of the Kyoto Encyclopedia of Genes and Genomes (KEGG, https://www.genome.jp/kegg/, assessed on 1 March 2024) orthologs (KOs) that represent genes in different species that typically retain the same function was obtained from PICRUSt2 analysis. The KOs involved in C fixation and N-cycling pathways were extracted for further analysis. The KEGG modules in C fixation and N-cycling pathways identified in the current study are presented in Table 1.

### 2.6. Statistical Analysis

All statistical analyses were performed in R (ver. 4.2.0) [53]. The alpha diversity indices are quantitative measures representing the diversity of ASVs in a sample. The Shannon Index (H), Simpson Index (D), and Chao1 Index were calculated using the vegan (version 2.6-6.1) [54] and biodiversityR (version 2.16-1) [55] packages. The Shannon-based True Diversity (Shannon-TD = exp (H)) and the Simpson-based True Diversity (Simpson-TD = 1/D) were calculated as suggested by Jost [56].

The alpha diversity indices were checked for normality using the *shapiro.test* function and were transformed as needed using the *boxcox* function in the MASS package (version 7.3-60.4) [57]. Linear mixed models were used to assess the main effects of tillage and rotation, current crop, soil depth, and their interaction effects on soil physicochemical properties, alpha diversity indices, and centered log ratio-transformed KOs (functional genes) using the *lme* function in the nlme package (version 3.1-164) [58] at a significance level of *p* ≤ 0.05. Tillage, rotation, current crop, and soil depth were treated as fixed effects, and block as a random effect. Multiple comparison was performed by Sidak adjustment using the *emmeans* function in the emmeans package (version 1.10.2) [59]. The *ACNOMBC* function in the ACNOMBC package was also used to assess the treatment effects on the relative abundances of taxa at the phylum and the genus levels [60].

The *adonis* function from the vegan package was used to perform permutational multivariate analysis of variance [61] to determine the main effects of tillage, rotation, current crop, soil depth, as well as their interaction effects on the heterogeneity of the bacterial communities. This analysis was based on robust Aitchison dissimilarity using the *vegdist* function in vegan [54]. Pairwise comparisons between treatments (tillage, rotation, or their combination) were conducted by using the *pairwise.adonis* function from the pairwiseAdonis package (version 0.4) [62] when one factor or interaction effect was significant. A distance-based redundancy analysis (dbRDA) was conducted to evaluate the correlation of soil physicochemical properties and soil management factors (tillage, crop, rotation, and soil depth) with soil bacterial community heterogeneity using the *capscale* function in vegan [54]. The correlation between KOs associated with C fixation and soil chemical properties were evaluated by Spearman’s rank-order correlation analysis using the *rcorr.adjust* function in the RcmdrMisc package (version 2.9-1) [63].

All bacterial taxon names at all ranks are written in italics as recommended by *Advancements of Microbiology* (http://am-online.org/preparing-manuscript/detailed-recommendations/, assessed on 1 March 2024).

## 3. Results

### 3.1. Soil Nutrient Content

This study investigated the main and interaction effects of tillage, crop rotation, and soil depth on soil nutrient content. Linear mixed-effect models revealed that there were significant interactions between tillage practices and soil depth. Notably, the soil total C and N concentrations were significantly higher at the 0–15 cm depth compared to the 15–30 cm depth under NT, whereas there were no such differences under CT. This difference may be ascribed to the crop residues left on the soil surface under NT, which were ploughed into the soil under CT. Consequently, at the 15–30 cm depth, total C and total N were higher in the CT plots compared to the NT plots, while at the 0–15 cm depth, the tillage practices did not significantly impact these parameters (Table 2). Neither crop rotation nor the current crop type significantly affected total soil C or N at either soil depth. The available phosphorus levels remained consistent across treatments in the 0–15 cm soil layer. However, the available potassium level was significantly higher under NT than CT at the same depth, suggesting a potential depth-dependent distribution of nutrients in response to different tillage practices. The soil moisture content was significantly greater under CT than under NT at the 15–30 cm depth but not at the 0–15 cm soil depth, likely due to enhanced deeper water penetration caused by CT. The soil pH remained unaffected by tillage at both soil depths.

### 3.2. Soil Bacterial Community Diversity and Compositional Structure

The present study investigated the changes in the soil microbial communities at two soil depths under different tillage and crop rotation regimes. The final ASV abundance table of the bacterial communities contains 2,820,458 high-quality reads, with 29,380 ± 5,261 (MEAN ± SD) reads per sample. In total, 37,378 ASVs were identified across all samples, spanning 65 phyla, 147 classes, 379 orders, 604 families, and 1144 genera. The soil bacterial communities were dominated by *Proteobacteria*, *Actinobacteriota*, and *Acidobacteriota*, accounting for 28.9%, 21.7%, and 17.6% of the total abundance, respectively. The most abundant genera recovered (≥2%) were all Candidatus groups, including *WHSN01* (in *Vicinamibacterales*) (2.60 ± 0.62%), *VBCG01* (in *Casimicrobiaceae*) (2.41 ± 1.01%), *GWC2-73-18* (in *Limnocylindrales*) (2.38 ± 0.51%), and *GMQP-bins7* (in *Gaiellaceae*) (2.29 ± 0.75%) (Appendix A).

The alpha diversity of the soil bacterial community, represented by Simpson index-based true diversity (Simpson-TD), Shannon–Weiner index-based true diversity (Shannon-TD), and Chao1 index, was significantly affected by the interaction between tillage practices and soil depth. Notably, these diversity indices were higher under CT than under NT at the 15–30 cm depth; however, no such difference was observed at the 0–15 cm depth. Additionally, under CT, the diversity indices were greater at the 15–30 cm depth compared to the 0–15 cm depth (Figure 1A). Furthermore, at the 0–15 cm soil depth, we observed higher alpha diversity (Simpson-TD and Shannon-TD) associated with corn compared to soybean under CT but not under NT (Figure 1B).

The compositional structure of the soil bacterial communities varied significantly with tillage practices (*p* = 0.001, F = 2.128) and their interaction with soil depth (*p* = 0.001, F = 1.508, Figure 2A). The crop type had a significant effect on the community compositional structure under CT (*p* = 0.02, F = 1.10), but this effect was not observed under NT (*p* = 0.40, F = 1.0). Notably, the differences were significant between wheat and corn (*p* = 0.05) and between wheat and soybean (*p* = 0.038) under CT (Figure 2B). The changes in community structure were not statistically significant (*p* = 0.451, F = 0.992) in response to the rotation regimes only. Distance-based RDA analysis suggested that the community structure was significantly affected by soil pH, soil total C and total N, and the C/N ratio (Figure 2C).

The tillage practices notably influenced the distribution of certain bacterial phyla. For example, at the 0–15 cm depth, six phyla were more abundant under CT than under NT (Figure 2D), while at the 15–30 cm depth, twelve phyla exhibited higher abundance under CT than under NT (Figure 2E). Furthermore, under NT, *Proteobacteria*, *Verrucomicrobiota*, and *Cyanobacteria* were more abundant at the 0–15 cm depth compared to the 15–30 cm depth, while *Chloroflexota*, *Firmicutes_D*, *Gemmatimonadota*, *Methylomirabiolata*, and *Desulfobacterota_B* were more abundant at the 15–30 cm depth than at the 0–15 cm depth (*p* ≤ 0.01, Figure 2F). No bacterial phyla showed stratified effects under CT (*p* = 0.600).

### 3.3. Microbial Carbon Fixation

One objective of this study was to evaluate the effects of tillage and crop rotation on energy metabolism processes related to C fixation in the Calvin–Benson cycle (map00710) and other C fixation pathways (map00720) at different soil depths. In this study, PICURSt2 predicted 115 KEGG orthologs (KOs) involved in C fixation, with 19 KOs at the 0–15 cm depth and 36 KOs at the 15–30 cm depth that differed significantly between CT and NT (Figure 3A). Interestingly, the majority of these KOs (81.6% at the 0–15 cm depth and 73.2% at the 15–30 cm depth) were more abundant under NT than under CT (Figure 3A, Appendix A).

No-till (NT) resulted in a depth-dependent enrichment of KOs across key metabolic pathways. Specifically, the number of KOs related to the reductive citric acid cycle (M00173) increased from 10, at the 0–15 cm depth, to 17 at the 15–30 cm depth; those related to the hydroxypropionate–hydroxybutylate cycle (M00375) increased from 2 to 3; and those related to the Wood–Ljungdahl pathway (M00377) increased from 3 to 7. However, only a small number of KOs were depleted under NT at the 15–30 cm depth, with five in the reductive citric acid cycle (M00173) and four in the 3-hydroxypropionate bi-cycle (M00376). This pattern suggests a more pronounced microbial or biochemical response to the NT treatment with an increasing soil depth. PICRUSt2 identified *Nitrospira_C*, *Solirubrobacter*, and *GMQP-bins7* (in *Gaiellaceae*) as major contributors to the tillage effect on KOs in the reductive citric acid cycle, while *DP-6* (in *Desulfobacterota*) predominantly affected the Wood–Ljungdahl pathway (Figure 3B). However, these taxa’s involvement in the stated C fixation pathways needs to be verified.

Additionally, specific genes showed positive correlations with total C, total N, pH, and available potassium (Appendix A). Three KOs (K01086, K01623, K01807) associated with the Calvin–Benson pathway (M00165) of C fixation were in significantly greater abundance for wheat compared to corn or soybean (Figure 3C). Specifically, *Piscinibacter* highly contributed to K01086 (*fbp-SEBP*) gene distribution.

### 3.4. Microbial Nitrogen Cycling

PICURSt2 identified 26 KOs associated with the N metabolism pathway (KEGG map00910), representing 18,397 ASVs. Among these KOs, most tillage-responsive genes were more abundant under NT compared to CT at both soil depths (Figure 4A,B). For example, KOs affiliated with the denitrification process (module M00529), including four at the 0–15 cm depth (K02305, *norC*; K02567, *napA*; K02568, *napB*; K15864, *nirS*) and two at the 15–30 cm depth (K02305, K15864), were more abundant under NT than under CT. Moreover, varying trends were observed for KOs involved in the assimilatory nitrate reduction process, like K00366 (*nirA*) and K00360 (*nasB*) (Figure 4A,B). In addition, K02586 (*nifD*) and K02591 (*nifK*) associated with N_2_ fixation (module M00175) were more abundant under NT compared to CT only at the 0–15 cm soil depth.

In surface soils (0–15 cm), *Microviga*, *Skermanella*, *VBCG01* (in *Casimicrobiaceae*), and *SHXO01* (in *Burkholderiales_597441*) were major contributors to N_2_ fixation and/or denitrification processes. These genera exhibited higher abundances under NT compared to CT, with *Skermanella* showing a statistically significant increase (*p* < 0.05). Furthermore, compared to CT, NT significantly enriched *Solirubrobacter* at the 0–15 cm depth, which was found to be affiliated with the major assimilatory nitrate reduction process (K00366) (Figure 4C). At the 15–30 cm soil depth, NT significantly enriched many N-cycling bacteria; *Rhodoferax_A_585629*, *Rhizobium_A_501058*, *Desulfosporosinus*, and *Pararhizobium,* associated with K02305 (*norC*), were significantly more abundant under NT compared to CT (Figure 4D). However, *Streptomyces_G_399870* and *Nocardioides_A_392796* affiliated with K00360 (*nasB*) (Figure 4D) were significantly more abundant under CT than NT.

The crop type had a significant effect on five KOs (Figure 4E). Wheat production principally resulted in higher abundances of K00360 (assimilatory nitrate reduction), K02586 and K02591 (N_2_ fixation), and K00374 (denitrification) compared to corn and soybean productions. Notably, *Nocardiodes_A_392796* and *Streptomyces_G_399870* emerged as the major contributors to nitrate reductase; *Bradyrhizobium* significantly contributed to N_2_ fixation, with higher abundance in wheat than in corn productions, while *VBCG01* (in *Casimicrobiaceae*) and *Massillia* were major contributors to the denitrification process (Figure 4F).

## 4. Discussion

### 4.1. Tillage and Depth Effects on Overall Soil Microbial Communities

The current study demonstrates that 18 years of tillage had a more substantial effect on the soil bacterial community and its involvement in C and N cycling than crop rotation or crop in monoculture. These results are consistent with other studies highlighting tillage as the predominant agricultural practice, along with the presence of crop residue, influencing the bacterial community structure in various agroecosystems [5,9,64,65]. This study underscores the significant effect of the interaction between tillage and soil depth on bacterial diversity, which was only observed with NT but not with CT. Conventional tillage increased the community diversity at greater soil depths compared to NT, which may be attributed to higher levels of total C and total N resulting from residue accumulation in the lower soil profile due to CT ploughing [66]. However, no differences in soil bacterial diversity were observed between CT and NT treatments in surface soils, nor between surface and deeper soils in NT soils. These findings are in line with those of Li et al. [15] who found no significant differences in soil bacterial diversity between NT and CT at the 0–5 cm depth, but higher diversity under CT in deeper (5–20 cm depth) soils. Li et al. [15] also observed higher community compositional variation across different soil depths under NT than under CT. Similarly, Adeleke et al. [67] reported insignificant differences in community diversity between CT and NT in surface samples (0–15 cm depth) in dark clay soils. However, these results contrast with those of Dong et al. [68] and Cao et al. [16], who reported increased bacterial diversity under NT than under CT in surface soils (0–20 cm, 0–25 cm) across multiple soil types and cropping systems. Such discrepancies may be partly attributed to the soil type, as the heavier soils in the present study, with less aeration under NT, favor anaerobic bacteria, potentially reducing soil bacterial diversity [69].

NT enriched *Chloroflexota* in deeper soils (Figure 2F). Members of this phylum possess a versatile facultative anaerobic metabolism [70] allowing them to thrive in the deeper, less aerated layers where organic matter is less disturbed and more stratified, providing niches that are suitable for their growth. NT also enriched *Eisenbacteria* and *Krumholzibacteriota* at both soil depths. PICRUSt2 predicted the affiliation of 287 *RBG-16-71-46* ASVs (belonging to *Eisenbacteria*) with C fixation [71], providing compelling evidence for the capacity of NT to enhance this vital ecological process. The *Krumholzibacteriota* phylum, initially characterized by Campbell et al. [72], has genes for benzoyl-CoA dearomatization, along with other genes governing the anaerobic catabolism of monoaromatics and functioning in the breakdown of recalcitrant C. By contrast, CT enriched *Proteobacteria*, a dominant soil bacterial phylum involved in the global cycling of C, N, and sulfur [73,74], at the 15–30 cm soil depth. This was likely due to higher total soil C and N levels in deeper soils resulting from CT, supporting the oligotrophic–copiotrophic theory [75], which predicts that copiotrophic taxa are more associated with greater labile C pools and flourish in soils with higher net C mineralization rates. *Cyanobacteria* and *Fibrobacterota* were also more abundant under CT, suggesting that these phyla favor the greater oxygen inputs and residue incorporation typical under CT [76].

### 4.2. Tillage and Depth Effects on Microbial C Fixation

Autotrophic bacteria play a crucial role in atmospheric carbon dioxide (CO_2_) assimilation and fixation and thus are important players in soil organic matter sequestration [33,77,78,79,80,81]. This study investigated the tillage and depth effects on the six known CO_2_ fixation pathways [78], such as the Calvin–Benson and the reductive citric acid cycles, in soil microbial communities. Ge et al. [33] reported that CT decreased the abundance of *cbbL*-carrying bacteria involved in the Calvin–Benson cycle compared to NT, using a quantitative PCR method. The present study, however, showed an insignificant influence on this metabolism pathway by the tillage practice (NT vs. CT, Appendix A). This limited tillage effect may be attributed to the dominance of the Calvin–Benson cycle in nutrient-rich environments [82,83,84], and both CT and NT soils in this study are considered nutrient-rich (Table 1). In contrast, the current study found that NT enriched bacteria that are involved in the reductive citric acid and other alternative C fixation pathways [85], such as *Nitrospira_C*, *Solirubrobacter*, and *GMQP-bins7* (in *Gallionellaceae*) (Figure 3A). These bacteria are particularly energy-efficient and can thrive in microanaerobic conditions [86]. Indeed, NT practices uphold the soil structure, reduce disturbance, and potentially stimulate microanaerobic zones by restricting oxygen diffusion [87,88]. NT also enriched *DP-6* (in *Desulfobacterota*) at both soil depths, which uses the Wood–Ljungdahl pathway (K15023, K00194, and K14138, Appendix A) for C fixation [85]. In contrast, CT increases aeration and oxygen penetration, creating a more aerobic environment in deep soils [76,89]. Additionally, the NT treatment resulted in a depth-dependent enrichment of KOs across key C fixation pathways, with a more pronounced microbial response to the NT treatment at a greater soil depth. This stratified response indicates that NT fosters distinct microbial communities and functions at different soil depths, enhancing nutrient-cycling efficiency and soil health. It is worth noting that there were no significant differences in total C content between CT and NT in surface soils. Consequently, the abundance of most bacterial communities and genes involved in C fixation showed no significant correlation with total C. Such an observation suggests that specific fractions of C (e.g., dissolved organic C), rather than total C content, may be the drivers for microbial C fixation under NT.

In summary, the present study highlights the positive impact of NT practices on soil bacterial communities involved in C fixation, suggesting that adopting NT could be a sustainable strategy to enhance microbial activity for C sequestration.

### 4.3. Tillage and Depth Effects on Microbial N Cycling

The present study showed that the impact of the tillage practices on microbial N cycling was dependent on the soil depth. The long-term adoption of NT significantly enriched genes involved in N_2_ fixation, assimilatory nitrate reduction, and denitrification, particularly in surface soils (0–15 cm depth). Similar findings were reported by Zhang et al. [42] in a corn monoculture system on sandy soils. For example, in surface soils, NT enriched *Solirubrobacter* and promoted nitrate assimilation, as indicated by the elevated abundance of K00366 (*nirA*) [90], compared to CT. This finding aligns with those of Yang et al. [91], who identified *Solirubrobacter* as a key player in maintaining the functionality of the N cycle across diverse land use types on the Tibetan Plateau [90]. Previous studies also highlighted the capacity of *Solirubrobacter* to degrade complex N-containing substrates such as chitin [92,93]. Conversely, at the 15–30 cm depth, CT enriched *Nocardioides* and *Streptomyces* and facilitated the reduction of nitrite to ammonium pathway (K00360, *nasB*) [90] compared to NT. These results indicate varying microbial responses to tillage practices across soil depths, emphasizing the need for a comprehensive understanding of the microbial dynamics involved in N cycling under varying environmental conditions.

It is well documented that in various legume production systems [94,95], no-till (NT) promoted N_2_ fixation, exemplified by increased nodulation [96,97], a higher proportion of plant N derived from atmospheric N_2_, and increased grain yield [98]. In our study, at the 0–15 cm soil depth, NT enriched the *nifD* (K02586) and *nifK* (K02591) genes, promoting the process of fixing N_2_ to ammonia, compared to CT. This effect was partially attributed to the enrichment of *Skermanella* under NT. Previous studies identified *Skermanella* as the most abundant diazotrophic genus in the soils of cucumber cropping systems, with its abundance positively correlated with nitrate levels [99].

Tillage also affected the microbial denitrification process and genes encoding nitrate reductase (*nar*, *nap*, or *nas*), nitrite reductase (*nir*), NO reductase (*nor*), and N_2_O reductase (*nos*) [100,101]. Among these, nitrite reductase (*nir*) is recognized as a principal source of soil N_2_O, while N_2_O reductase (*nos*) is recognized as a principal sink of soil N_2_O [102,103]. NT enriched *napA* and *napB* (involved in the transformation from nitrate to nitrite), *nirS* (regulating the process from nitrite to nitric oxide), and *norC* (regulating the process from nitric oxide to nitrous oxide) compared to CT at the 0–15 cm soil depth. At the 15–30 cm depth, NT also increased the abundance of *norC* and *nirS* compared to CT. Our study suggests a potential for greater N_2_O emission under NT compared to CT, which aligns with previous studies reporting higher N_2_O emission under NT in eastern Canada [104] and Quebec [105], where higher soil density and moisture and lower oxygen levels prevail [104].

In summary, our study found that NT promoted both microbial N_2_ fixation and denitrification, which differentially contribute to soil N storage. Determining the dominant process requires further investigation. Additionally, further research on greenhouse gas emissions is essential to validate our observations and understand the complex interactions involved.

### 4.4. Crop Rotation Effects on Soil Bacterial Community Diversity, Structure, and Functions

In comparison to the tillage practices, crop rotation and current crop appeared to have limited effects on soil bacterial community diversity and compositional structure in the current study. No marked impact of the crop rotational regimes on the soil bacterial community was observed, consistent with the results of Navarro-Noya et al. [64]. This may be attributed to an insignificant response of the soil physicochemical properties to crop rotation. Contrary to our findings, Chamberlain et al. [106] found that the soil bacterial community compositional structures in the corn and soybean phases of an annual rotation were similar, but distinct from those in continuously cropped corn or soybean. Soman et al. [107] also reported marked differences in the soil bacterial community structure of corn across different rotation regimes. These disparities suggest that a combination of crop rotation and tillage practices can influence the soil bacterial communities in a unique way, adding complexity to our understanding of soil microbial ecology in agroecosystems.

Our study revealed that the crop types, rather than rotation sequences, significantly influenced specific soil bacterial communities associated with C fixation and N cycling, particularly when comparing wheat to corn and soybean. Wheat specifically promoted microbial activity related to the Calvin–Benson cycle of the C fixation pathway, as well as key N-cycling processes, such as N_2_ fixation, assimilatory nitrate reduction, and denitrification, compared to corn and soybean. For instance, in the present study, wheat increased the abundance of *Piscinibacter* and *Bradyrhizobium* in comparison with corn. *Piscinibacter* is a facultative methylotrophic bacterium that can utilize single-carbon compounds, which highlights its capability in carbon fixation, particularly through the Calvin–Benson cycle [108]. *Bradyrhizobium* spp. were reported to play a major role in N_2_ fixation, especially in legume-associated systems [109]. These noteworthy effects of crop types are likely attributed to crop-induced alterations in resource availability, stemming from variations in root exudates and crop residue quality [20,110,111].

The photosynthetic pathways of these crops revealed that wheat, a C3 monocot, displayed unique characteristics in shaping soil bacterial communities, when compared to corn, a C4 monocot, and soybean, a C3 dicot. Despite both wheat and soybean being C3 plants, soybean did not exhibit the same distinct promotion of bacterial genes involved in the Calvin–Benson cycle as wheat. This discrepancy may be attributed to differences in their physiological and biochemical traits beyond their photosynthetic pathways. Wheat’s root exudates might contain specific compounds that selectively promote microbes involved in the Calvin–Benson cycle and N-cycling processes [112]. In contrast, soybean has a symbiotic relationship with N_2_-fixing bacteria in its root nodules, which harbor different microbial community dynamics compared to wheat [113,114]. Additionally, the quality and decomposition rates of soybean residues might provide different supplies of nutrients from wheat residues, influencing the soil microenvironment and microbial community structure differently [20,115]. Understanding these crop-specific interactions is crucial for optimizing crop management practices to enhance soil health and nutrient cycling efficiency.

## 5. Conclusions

The present study demonstrated that tillage practices significantly shaped soil bacterial community diversity, structure, and functions related to C fixation and N cycling more than rotation and crop type. The stratification of bacterial communities was observed under NT conditions but not under CT. NT enhanced microbial functions associated with C and N cycling, especially through the reductive citric acid cycle and denitrification at both soil depths and N_2_ fixation at the 0–15 cm depth. The crop type also influenced the soil bacterial communities involved in C sequestration and N cycling. These findings indicate that the long-term tillage practices had a greater influence than crop rotation on the soil bacterial community, affecting its diversity, compositional structure, and functionality, particularly in C- and N-cycling processes. This study highlights the importance of integrated management practices for sustainable agriculture. Even though this study was based on a snapshot evaluation after an 18-year long-term experiment, this one-year study provides valuable insights into the impact of agricultural practices on the soil microbiome. Further investigation over time will be conducted to validate our findings.

## Figures and Tables

**Figure 1 microorganisms-12-01635-f001:**
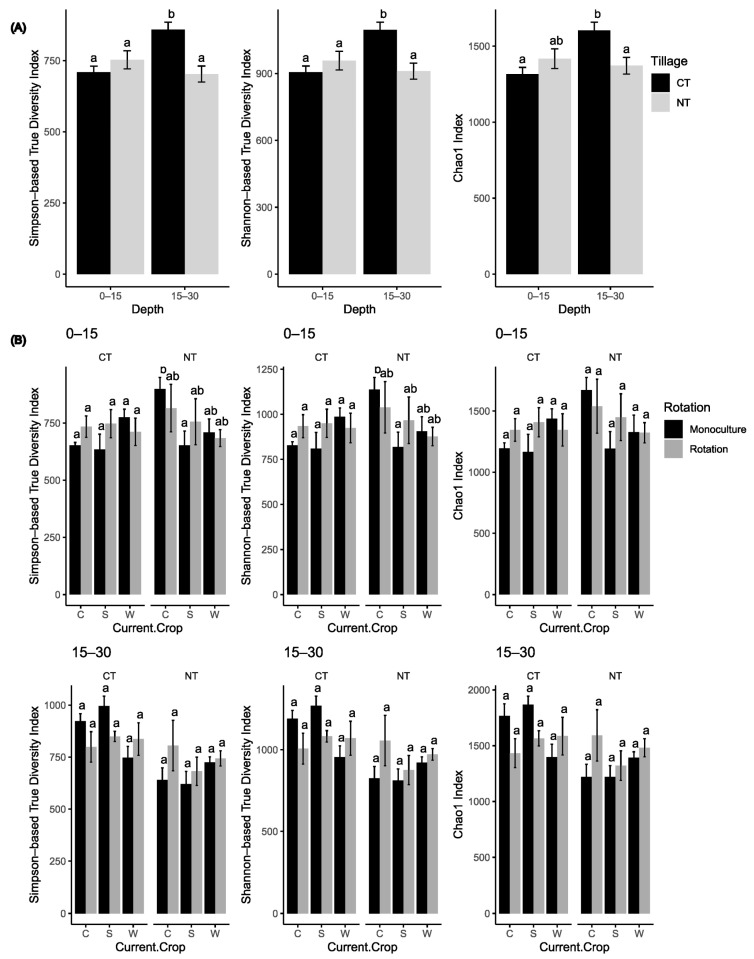
Soil bacterial community diversity indices affected by the interaction of tillage and soil depth (**A**) and the interaction of tillage, rotation, and crops at 0–15 cm and 15–30 cm soil depths (**B**). CT, conventional tillage; NT, no-till; C, corn; S, soybean; W, wheat. Error bars represent standard errors. Different letters across all treatments in panel A and across all treatments under CT and NT in panel B represent significant differences at α = 0.05 according to Sidak adjustments.

**Figure 2 microorganisms-12-01635-f002:**
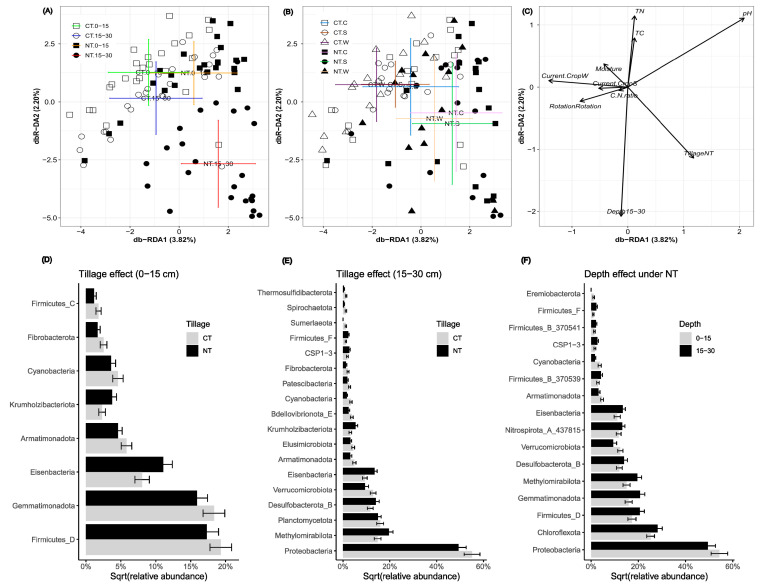
Soil bacterial community structure affected by tillage, rotation, crop type, soil depths, and soil physicochemical properties determined by db-RDA. Soil bacterial community structure affected by the interaction of tillage and soil depth (**A**), the interaction of tillage and crops (**B**), and all factors (**C**), and soil bacterial phyla affected by tillage at 0–15 cm (**D**) and 15–30 cm soil depths (**E**) and by depth under NT (**F**). CT, conventional tillage; NT, no-till; 0–15, 0–15 cm soil depth; 15–30, 15–30 cm soil depth. Error bars represent standard errors.

**Figure 3 microorganisms-12-01635-f003:**
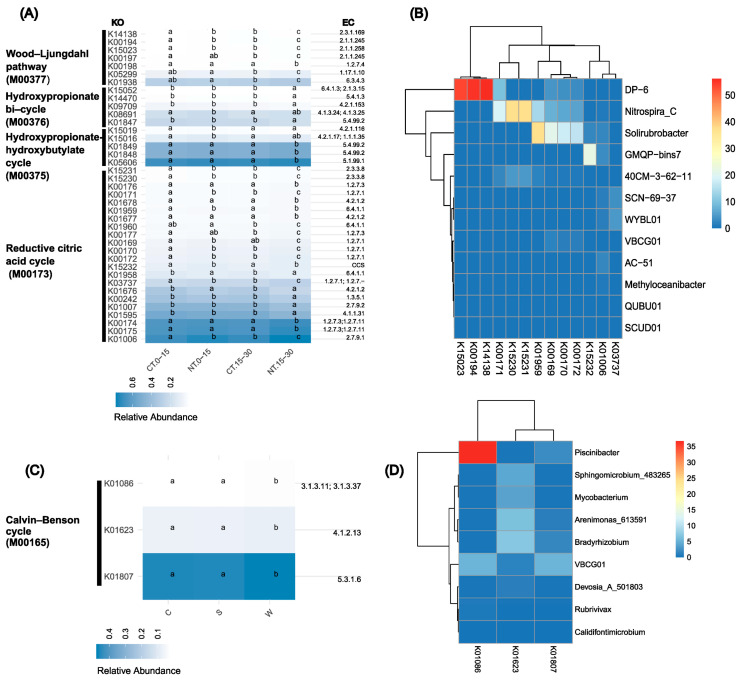
Functional genes associated with carbon fixation, as predicted by PICRUSt2. The relative abundances of KEGG orthologs (KOs) that were significantly influenced by the tillage practices at either 0–15 cm or 15–30 cm soil depth within four key modules from map00720 (other C fixation pathways except Calvin–Benson cycle) are shown, including the reductive citric acid cycle (M00173), the hydroxypropionate–hydroxybutylate cycle (M00375), the 3-hydroxypropionate bi-cycle (M00376), and the Wood–Ljungdahl pathway (M00377) (**A**). Additionally, KOs within the Calvin–Benson cycle module (M00165) were significantly influenced by the crop types (**C**). The corresponding enzyme IDs (EC) for each KO are provided on the right side of the panels. Different letters across all treatments in panels (**A**,**C**) represent significant differences at α = 0.05 according to Sidak adjustments. Panels (**B**,**D**) display heatmaps showing the abundance of bacterial genera significantly affected by tillage (**B**) or crop types (**D**), which importantly contributed to the KOs presented in panels (**A**,**C**), respectively. Colors and density represent the proportion (%) of a specific gene’s abundance within various genera, with blue indicating lower abundance, and red indicating higher abundance. CT, conventional tillage; NT, no-till; 0–15, 0–15 cm soil depth; 15–30, 15–30 cm soil depth. Crop types: C for corn, S for soybean, and W for wheat.

**Figure 4 microorganisms-12-01635-f004:**
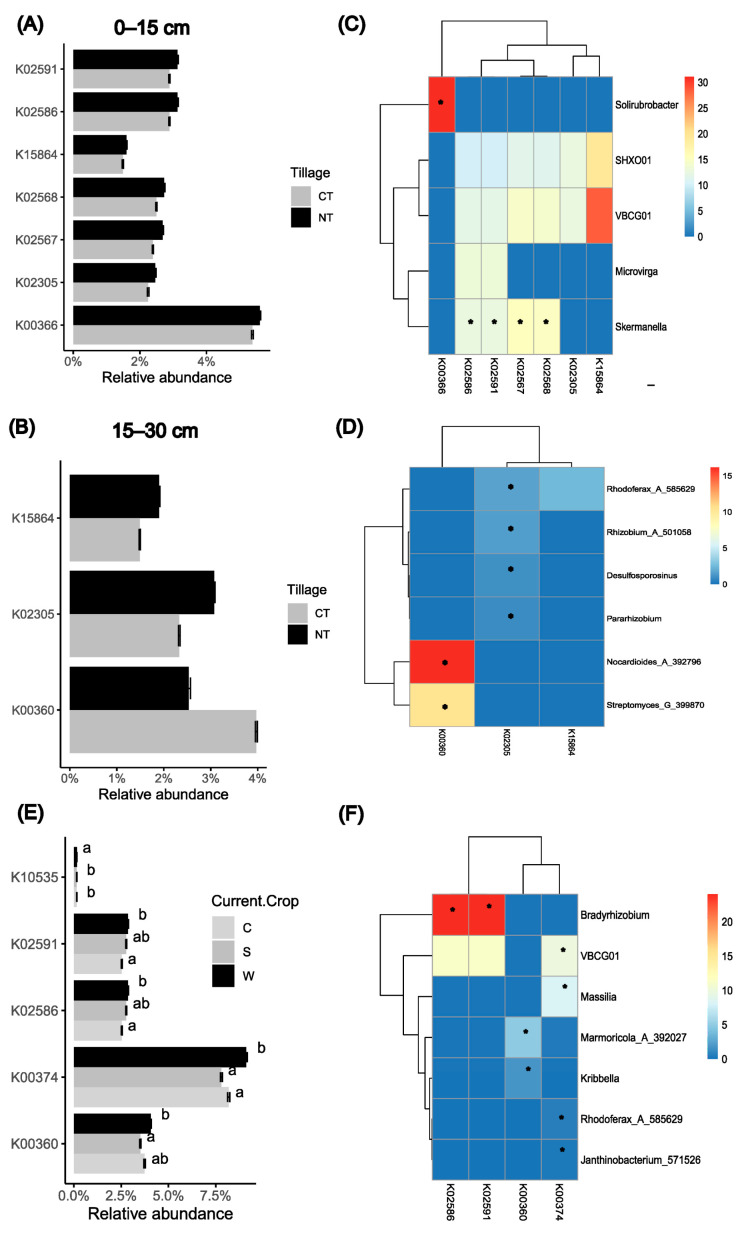
Functional genes associated with nitrogen (N) metabolism, as predicted by PICRUSt2. Panels (**A**,**B**,**E**) illustrate the relative abundance of KEGG orthologs (KOs) significantly influenced by the tillage practices at 0–15 cm soil depth and 15–30 cm soil depth and the current crop, respectively. Different letters (a, b, c) above the bars in panel (**E**) represent significant differences between the crops at α = 0.05 according to Sidak adjustments. Results for crops sharing the same letter are not significantly different from each other, while those for crops with different letters are significantly different. Panels (**C**,**D**,**F**) display heatmaps showing the abundance of bacterial genera that significantly contributed to the KOs affected by the tillage practices at 0–15 cm soil depth (**C**) and 15–30 cm soil depth (**D**) and the current crop (**F**). Stars in the heatmaps indicate significant differences in the abundance of genera between conventional tillage (CT) and no-till (NT) in panels (**C**,**D**), and higher abundance in wheat compared to corn or soybean in panel (**F**) (*, *p* ≤ 0.05). Different colors represent the proportion (%) of a specific gene’s abundance within various genera. CT, conventional tillage; NT, no-till; C, corn; S, soybean; W, wheat. Error bars represent standard errors.

**Table 1 microorganisms-12-01635-t001:** KEGG modules in C fixation and N-cycling pathways identified in the current study.

Module	Description	Pathway Map
Carbon fixation		
M00173	Reductive citrate cycle (Arnon–Buchanan cycle)	map00720
M00374	Dicarboxylate–hydroxybutyrate cycle	map00720
M00375	Hydroxypropionate–hydroxybutylate cycle	map00720
M00376	3-Hydroxypropionate bi-cycle	map00720
M00377	Reductive acetyl-CoA pathway (Wood–Ljungdahl pathway)	map00720
M00165	Reductive pentose phosphate cycle (Calvin–Benson cycle)	map00710
Nitrogen metabolism		
M00175	Nitrogen fixation, nitrogen => ammonia	map00910
M00528	Nitrification, ammonia => nitrite	map00910
M00529	Denitrification, nitrate => nitrogen	map00910
M00530	Dissimilatory nitrate reduction, nitrate => ammonia	map00910
M00531	Assimilatory nitrate reduction, nitrate => ammonia	map00910
M00804	Complete nitrification, comammox, ammonia => nitrite => nitrate	map00910
M00973	Anammox, nitrite + ammonia => nitrogen	map00910

**Table 2 microorganisms-12-01635-t002:** Effects of tillage, rotation, crop, and soil depth on total soil carbon (TC), nitrogen (TN), C/N ratio, soil pH, moisture, and available phosphorus (P) and potassium (K).

	TC	TN	C/N Ratio	pH	Moisture	Available P	Available K
	%				%	mg kg^−1^	
0–15 cm							
CT ^1^	1.99 ± 0.97 a ^3^	0.15 ± 0.05 a	13.12 ± 1.54 a	7.04 ± 0.39 a	21.8± 3.55 a	41.0 ± 18.1 a	72.5 ± 30.0 b
NT	2.03 ± 0.59 a	0.16 ± 0.03 a	12.94 ± 1.40 a	7.11 ± 0.35 a	19.9 ± 3.47 ab	49.0 ± 21.4 a	105.0 ± 29.5 a
15–30 cm							
CT	1.77 ± 0.80 a	0.13 ± 0.04 ab	12.92 ± 1.20 a	7.05 ± 0.41 a	22.1 ± 2.51 a	N/A ^2^	N/A
NT	1.34 ± 0.34 b	0.11 ± 0.02 b	12.27 ± 1.28 a	7.28 ± 0.26 a	19.1 ± 1.88 b	N/A	N/A
Analysis of variance (*p* values)							
Tillage (T)	ns *	ns	ns	ns	ns	ns	<0.001
Rotation (R)	ns	ns	ns	ns	ns	ns	ns
Crop (C)	ns	ns	ns	ns	ns	ns	ns
T × R	ns	ns	ns	ns	ns	ns	ns
T × C	ns	ns	ns	ns	ns	ns	ns
R × C	na	0.048	ns	ns	ns	ns	ns
T × R × C	ns	ns	ns	ns	ns	ns	ns
Depth (D)	<0.001	<0.001	ns	ns	<0.001	-	-
T × D	0.01	0.013	ns	ns	ns	-	-
T × R × D	ns	ns	ns	ns	ns	-	-
T × C × D	ns	ns	ns	ns	ns	-	-
R × C × D	ns	ns	ns	ns	ns	-	-
T × R × C × D	ns	ns	ns	ns	ns	-	-

*, significance at *p* = 0.05; ns, not significant. ^1^ CT, conventional tillage; NT, no-till. ^2^ N/A, data of available P and K at 15–30 cm soil depth are not available. ^3^ Means followed by the same letter in each column are not significantly different at α = 0.05 according to Sidak adjustment.

## Data Availability

The raw sequencing data in FASTQ format are accessible from the NCBI Sequence Read Archive under the BioProject accession number PRJNA1129531 (https://dataview.ncbi.nlm.nih.gov/object/PRJNA1129531 accessed on 1 August 2024) and the Biosample accessions SAMN42159406- SAMN42159501.

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
