# Peer review of "Stratified Effects of Tillage and Crop Rotations on Soil Microbes in Carbon and Nitrogen Cycles at Different Soil Depths in Long-Term Corn, Soybean, and Wheat Cultivation"

_microorganisms, 2024, doi:10.3390/microorganisms12081635_

Round 1

Reviewer 1 Report

Comments and Suggestions for Authors

microorganisms-3140528: Stratified Effects of Tillage and Crop Rotations on Soil Microbes in Carbon and Nitrogen Cycles at Different Soil Depths in Long-Term Corn, Soybean, and Wheat Cultivation.

The manuscript is very interesting and of great importance in agriculture and environmental protection. It deals with the effects of tillage and crop rotation on soil microorganisms at different depths involved in the carbon and nitrogen cycles. The paper is generally well written. However, minor changes are required in the work, namely:

1. Table 2 needs formatting. Also under the table should be explained, what the lowercase letters mean.

2. Also under Figure 1, the lowercase letters used (homogeneous groups) should be explained.

3. Figures 2, 3, 4, S1, S3 are not very legible. Please correct them.

4. In Figure S4 there are very large standard deviations. Please provide homogeneous groups.

Reviewer 2 Report

Comments and Suggestions for Authors

In the manuscript submitted to Microorganisms, Wen Chen et al. present the results of extensive experiments comparing the effects of crop rotation and tillage on the soil microbiome (corn, soybean, and wheat are of great importance to the agricultural industry).

The abstract and introduction provide all the necessary introductory information for the reader.

Minor problems

1) Some typing errors and sentences that are difficult to interpret need editorial correction.

2) Chapter "2.7. Availability of sequence data" is empty

3) Some comments are related to the Tables and Figures

3.1) What do a and b mean in Table 2 and Figure 1 (also ab)?

3.2) What are the differences between Firmiculites C, D, F in Figure 2? What is CSP1-3?

3.3) In Figure 3 (B) the red area is prominent, what is its physiological significance? A similar question for Figure 3 (D). What do DP-6, QUBU01, and other names mean?

3.4) In Fig. 4 (D) the unclassified lower family zone is highlighted in red, what kind of bacteria are they? The red zone in Figure 4 (F) is also worth discussing. What do a, b, ab mean in Figure 4 (E)?

4) The Discussion and Conclusions section should be supplemented with a discussion of the physiological significance of the observed differences in the microbiome.

Comments on the Quality of English Language

Some typos and sentences that are difficult to interpret need editorial correction.

Reviewer 3 Report

Comments and Suggestions for Authors

Abstract part provides o good and comprehensive overview of the topic and background.

line 24  -  Instead of

 biological nitrogen (N2) fixation-related bacteria at 0-15 cm and denitrification-related bacteria

Should be

bacteria involved in biological nitrogen (N2) fixation and denitrification at 0-15 cm

Introduction part provides o good and comprehensive overview of the topic and background.

line 53 variability in bacterial community structure across soil depths compared to NT practices. Li, He, Guo, Zhang  and Li [15] – should be Li et al. (name first author only)

Line 60 - Scientific Latin names of plants and bacteria should be written in italic form!

                I n such a way throughout the manuscript.

 Lines 79-81- While numerous studies have explored the impact of tillage practices and crop rotation regimes on the soil microbiome, there is a need for further investigation into the ecological functions of specific microbial communities - which studies are they  please provide references

 Line 116- Using this study site - this should be deleted, it is superfluous

The methods are appropriate, and they been correctly described and applied.

The Results part is well structured and contains all results, which are clearly described in figures and tables.

Line 280 should edit table 2  technically

Line 309 Figure 1 (B). C, corn; CT, conventional tillage; NT, no-till; S, soybean; W, wheat. Error bars rep-308 resent standard errors.

(B). CT, conventional tillage; NT, no-till; C, corn; S, soybean; W, wheat. Error bars rep-308 resent standard errors.

In Discussion authors are describe and analyzes their results and linked them with already published literature.

In Conclusions the authors showed the  importance of integrated management practices to enhance productivity in agriculture.

Line 567 - It should not write  our study, please.

Lines 573 - 575 - These findings highlight the  importance of integrated management practices to enhance productivity in agricultural landscapes.

Landscapes – should be deleted

These findings highlight the  importance of integrated management practices to enhance productivity in agriculture.

Reviewer 4 Report

Comments and Suggestions for Authors

The article with the title „Stratified Effects of Tillage and Crop Rotations on Soil Microbes in Carbon and Nitrogen Cycles at Different Soil Depths in Long-Term Corn, Soybean, and Wheat Cultivation” is well written with interesting subject treated in details.

Some suggestions to manuscript improvement:

The abstract seems too long. Also make all text impersonal row 29 instead of we conclude you can say It was concluded or in conclusion, same for row 32.

Rows 60-61: Usually scientific names should be written in italics please see (Zea mays L.)- soybean (Glycine max L. Merr.)-wheat (Triticum anestivum L.)

Row 114: Please correct the text

Row 128 CaCl2, please write correct the chemical name also in row 151

Table 2 should be kept in a singular page; in this form it is difficult to follow the results

In figure 3 caption some results are written, please place the results in the text and the figure caption only for explaining the abbreviations used in the combined figures. Also why the figures have different quality, some writings are with low quality

The discussion section needs some improvement in case of the writing style see rows 511-520

The conclusion section must be rewritten in an impersonal style without our study and we observed. Also please highlight the conclusions about all parameters assessed, not only general aspect.
